# Synapsin E-domain is essential for α-synuclein function

**Alexandra Stavsky[1†], Leonardo A Parra-Rivas[2,3†], Shani Tal[1], Jen Riba[1], Kayalvizhi Madhivanan[2‡], Subhojit Roy[2,3,4]\*, Daniel Gitler[1]\***

[1]Department of Physiology and Cell Biology, Faculty of Health Sciences and School of Brain Sciences and Cognition, Ben-Gurion University of the Negev, Beer Sheva, Israel; [2]Department of Pathology, University of California, San Diego, La Jolla, United States; [3]Aligning Science Across Parkinson's (ASAP) Collaborative Research Network, Chevy Chase, United States; [4]Department of Neurosciences, University of California, San Diego, La Jolla, United States

**Abstract** The cytosolic proteins synucleins and synapsins are thought to play cooperative roles in regulating synaptic vesicle (SV) recycling, but mechanistic insight is lacking. Here, we identify the synapsin E-domain as an essential functional binding-partner of α-synuclein (α-syn). Synapsin E-domain allows α-syn functionality, binds to α-syn, and is necessary and sufficient for enabling effects of α-syn at synapses of cultured mouse hippocampal neurons. Together with previous studies implicating the E-domain in clustering SVs, our experiments advocate a cooperative role for these two proteins in maintaining physiologic SV clusters.

## eLife assessment

Alpha-synuclein is a synaptic vesicle associated protein that is linked to a number of neurodegenerative disorders. In this manuscript, the authors provide **compelling** evidence of alpha-synuclein's interaction with E-domain synapsins as the main culprit mediating the suppression of neurotransmitter release and synaptic vesicle recycling by alpha-synuclein. This **important** work provides molecular mechanisms underlying alpha-synuclein functions.

## Introduction

Substantial evidence links the small presynaptic protein α-syn to neurodegenerative diseases, collectively called synucleinopathies. The normal function of α-syn has been investigated for over a decade, and a prevailing view is that α-syn is a physiologic attenuator of neurotransmitter release. Modest overexpression of α-syn dampens synaptic responses (*Nemani et al., 2010*; *Scott et al., 2010*; *Wang et al., 2014*; *Sun et al., 2019*; *Atias et al., 2019*), and analogously, eliminating α-syn leads to phenotypes consistent with augmented synaptic release (*Abeliovich et al., 2000*; *Yavich et al., 2004*; *Yavich et al., 2006*; *Senior et al., 2008*; *Greten-Harrison et al., 2010*; *Anwar et al., 2011*), although the latter has not been seen in all studies (*Burré et al., 2010*). At a cellular level, synaptic attenuation is likely mediated by effects of α-syn on vesicle organization and trafficking, which are even seen in minimal in vitro systems, where recombinant α-syn clusters small synaptic-like vesicles (*Diao et al., 2013*; *Sun et al., 2019*). An emerging model is that α-syn plays a role in the organization and mobilization of SVs, that in turn regulates SV-recycling and neurotransmitter release; however, underlying mechanisms are unknown.

Work over several decades has shown that temporal and spatial regulation of the SV cycle is achieved by the cooperative effort of diverse groups of proteins, such as Muncs/SNAREs – orchestrating SV

**\*For correspondence:**
sroy@ucsd.edu (SR);
gitler@bgu.ac.il (DG)

†These authors contributed equally to this work

**Present address:** ‡Arrowhead Pharmaceuticals, Pasadena, United States

**Competing interest:** The authors declare that no competing interests exist.

docking, priming, fusion – and sequential assembly of a variety of endocytosis-related proteins that build a platform for efficient membrane retrieval. Reasoning that an understanding of functional α-syn partners would offer meaningful insight into α-syn function, we have been combining SV-recycling assays with structure-function approaches to identify the protein-network in which α-syn operates at the synapse. Using this approach, we recently found that the physiologic effects of α-syn at the synapse requires synapsins (*Atias et al., 2019*). While modest over-expression of α-syn in wild-type (WT) cultured hippocampal neurons attenuated SV recycling, there was no effect in neurons lacking all synapsins, indicating that synapsins were necessary to enable α-syn functionality. Reintroduction of the canonical synapsin isoform (synapsin Ia) reinstated α-syn mediated attenuation, confirming functional cooperation between α-syn and synapsins (*Atias et al., 2019*).

## Results

Synapsins are a family of cytosolic proteins with known roles in maintaining physiologic SV clusters (*Cesca et al., 2010*; *Orenbuch et al., 2012*; *Zhang and Augustine, 2021*), and recent work supports a model where SVs are confined within synapsin-based protein condensates (*Milovanovic et al., 2018*). Alternative splicing of three synapsin genes gives five major isoforms. Both synapsins and synucleins are peripherally associated with SVs via the N-terminus, while C-terminal regions are more variable and structurally disordered (*Song and Augustine, 2023*). Depending on the isoform, the C-terminus of synapsin has two to three structurally distinct domains, and substantial evidence indicates that this domain-variability leads to isoform-specific functions (*Song and Augustine, 2015*; *Song and Augustine, 2023*). Reasoning that identifying the specific synapsin domain/isoform that bound to α-syn and facilitated α-syn function would offer mechanistic insight into α-syn biology, we systematically evaluated effects of each synapsin isoform in enabling the physiologic effects of α-syn. For these experiments, we used pHluorin assays that report exo/endocytic SV recycling. pHluorin is a pH-sensitive GFP that acts as a sensor for pH changes, and in our experiments, the probe is tagged to the transmembrane presynaptic protein synaptophysin, and targeted to the interior of SVs [called 'sypHy', see *Royle et al., 2008*]. In resting SVs, sypHy is quenched, as the pH is acidic (~5.5). However, upon stimulation, SVs fuse with the presynaptic plasma membrane, resulting in pH-neutralization and a concomitant rise in fluorescence, which is subsequently quenched as the vesicles are endocytosed and reacidified (*Figure 1A*). Fluorescence fluctuations in this assay are a measure of SV exo/endocytosis, and at the end of the experiment, all vesicles can be visualized by adding $NH_4Cl$ to the bath (alkalinization). As reported previously, overexpression of h-α-syn attenuated SV recycling in WT hippocampal cultured neurons, but there was no effect in neurons from mice lacking all synapsins – synapsin triple knockout or TKO mice (*Figure 1B–C*).

The synapsin family has five main isoforms, Ia Ib, IIa, IIb, and IIIa (*Figure 1D*). To determine synapsin isoforms that enable α-syn functionality, we overexpressed h-α-syn in cultured neurons from synapsin TKO mice and systematically reintroduced each synapsin isoform, with the goal of identifying synapsin isoforms that reinstated α-syn-induced synaptic attenuation (see plan in *Figure 1E*). Reintroduction of Synapsin Ia (containing domains A-E) in this setting reinstated α-syn functionality (*Figure 1F*, left-panel; these changes are due to altered exocytosis, see *Figure 1—figure supplement 1A–B*). However, interestingly, only synapsins Ia, IIa, and IIIa enabled h-α-syn-mediated synaptic attenuation, whereas synapsins Ib and IIb had no effect (*Figure 1F*, quantified in *Figure 1G*). These effects were likely due to exocytosis as noted above, as quantification of the fluorescence decay-kinetics – which is a measure of endocytosis – did not reveal any changes (*Figure 1—figure supplement 1C–E*). One prediction of the pHluorin experiments is that the synapsin isoforms that allow for α-syn functionality would also be the ones that bind to α-syn. To test this, we performed co-immunoprecipitation experiments in neuronal cell lines, where we co-transfected neuro-2a cells with myc-tagged h-α-syn and each synapsin isoform (fluorescent-tagged), immunoprecipitated the synapsin isoform, and determined amounts of co-immunoprecipitated α-syn by western blotting (schematic in *Figure 2A*). While synapsins Ia, IIa, and IIIa bound robustly to h-α-syn, binding of synapsins Ib and IIb was much lower (*Figure 2B*, quantified in *Figure 2C*). Taken together, these experiments indicate that only three synapsin isoforms (Ia, IIa, and IIIa) can robustly bind to α-syn and reinstate functional effects of α-syn in this setting. Since the E-domain, within the variable C-terminus, is common to these three synapsin isoforms – and absent in the others (see domain-structure in *Figure 2D*) – we reasoned that the E-domain was the bona fide α-syn binding-site, and also responsible for facilitating α-syn functionality.

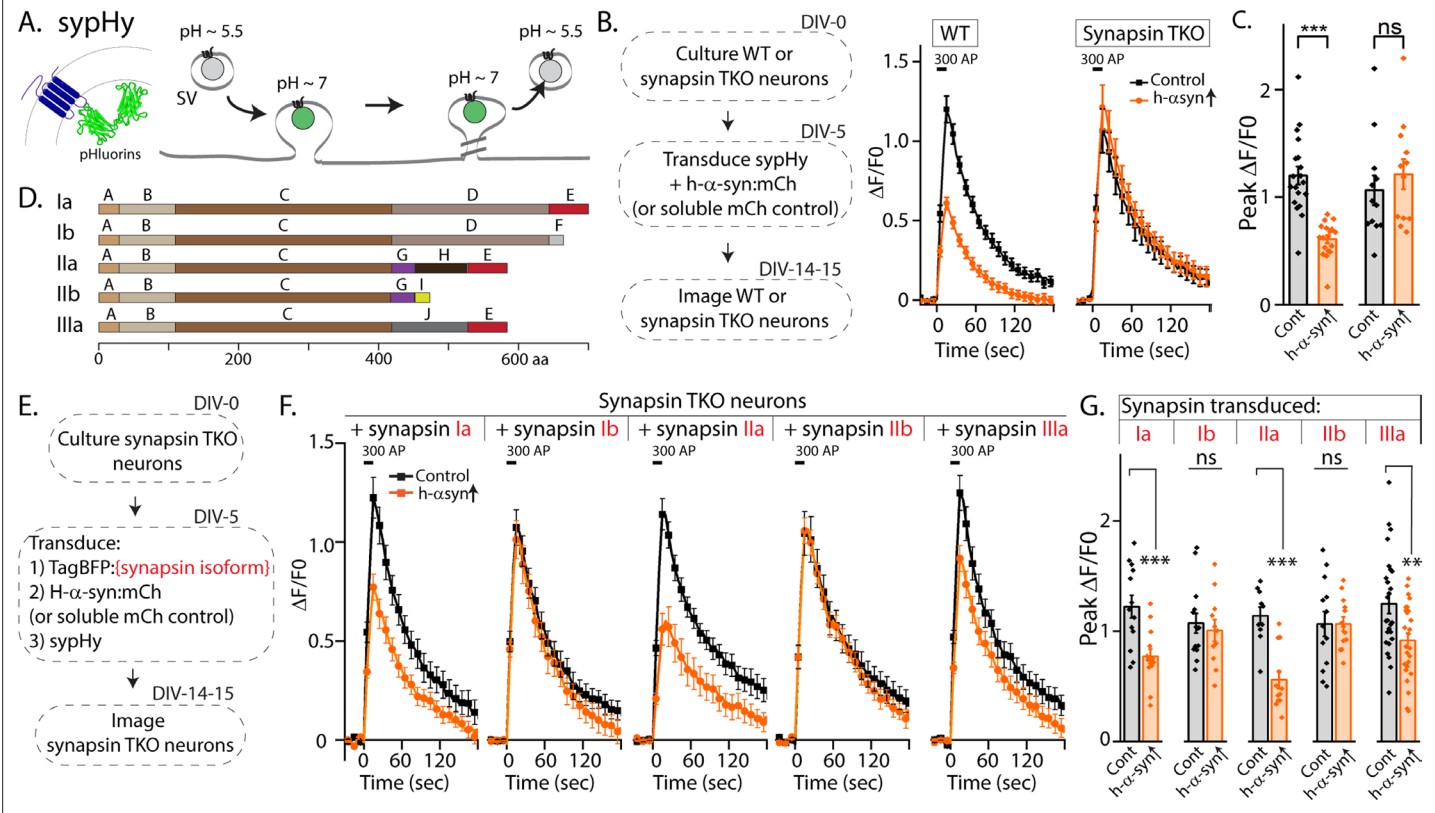

**Figure 1.** Screening for synapsin isoforms that allow α-syn functionality. (**A**) Schematic showing pH-sensitive sensor sypHy and principle of pHluorin experiments to quantitatively evaluate the SV cycle (see main text and methods for more details). (**B**) Elimination of all synapsins block α-syn functionality at synapses. Left: Schematic showing design of pHluorin experiments. WT or synapsin TKO cultured hippocampal neurons were co-transduced at 5 days in vitro (DIV) with h-α-syn:mCherry (or mCherry as control) and sypHy, and imaged at 14–15 DIV. Right: Stimulation-induced sypHy fluorescence traces (300 action potentials at 20 Hz, delivered at t=0 s – for clarity, symbols only mark every other mean ± SEM $\Delta F/F_0$ value in all sypHy traces). Note that while h-α-syn over-expression (orange) attenuated sypHy fluorescence in WT neurons, there was no effect in neurons from mice lacking all synapsins (TKO). All sypHy data quantified in (**C**). (**C**) Quantification of peak $\Delta F/F_0$ sypHy values (bars: mean ± SEM). A total of 12–19 coverslips were analyzed for each condition, from at least three separate cultures (***p=9e-7, ns p=0.45, student's t-test). (**D**) Domain structure of the five main synapsin isoforms. (**E**) Experimental design to identify the synapsin isoform that reinstated α-syn functionality, Synapsin TKO neurons were co-transduced at 5 DIV with each synapsin isoform, h-α-syn, and sypHy; and imaged at 14–15 DIV. (**F**) SypHy fluorescence traces (mean ± SEM). Note that h-α-syn(orange) attenuates SV recycling only if the neurons are also co-expressing the 'a' isoforms – synapsins Ia, IIa, and IIIa (300 action potentials at 20 Hz, delivered at t=0 sec). Data quantified in G. (**G**) Quantification of peak $\Delta F/F_0$ sypHy values (bars: mean ± SEM). 13–26 coverslips from at least three separate cultures were analyzed for each condition (from left to right: ***p=0.0009, ns p=0.62, ***p=0.00005, ns p=0.99, **p=0.004, student's t test).

The online version of this article includes the following source data and figure supplement(s) for figure 1:

**Source data 1.** Tabular data and statistical analyses for graphs presented in panels B, C, F, G.

**Figure supplement 1.** Effects of h-α-syn over-expression are largely due to suppression of exocytosis.

**Figure supplement 1—source data 1.** Tabular data and statistical analyses for graphs presented in panels A, B, D, and E.

In parallel experiments, we also narrowed down the reciprocal region in α-syn bound to synapsin. Toward this, we designed GST-pulldown assays to test the interaction of various h-α-syn sequences with mouse brain synapsins. In these experiments, beads with GST-tagged h-α-syn (WT, deletions, and scrambled variants) were incubated with mouse brain lysates, and brain synapsins binding to α-syn were evaluated by western blotting (***Figure 2E***). ***Figure 2F*** shows how the scrambled variants were designed. While synapsins bound to GST-tagged WT-h-α-syn, deletion of the C-terminus (α-syn 96–140) eliminated this interaction (***Figure 2G***, lanes 1–3). Regions within amino acids 96–110 of α-syn were critical in binding synapsin, as this minimal region bound to synapsin (***Figure 2G***, lanes 4–5), and scrambling the amino acids within this region – while keeping the other sequences intact – eliminated this interaction (***Figure 2G***, lanes 6–7). Data from all western blots is quantified in ***Figure 2G*** – bottom. Together, these experiments identify amino-acids 96–110 of α-syn as the region binding to synapsin.

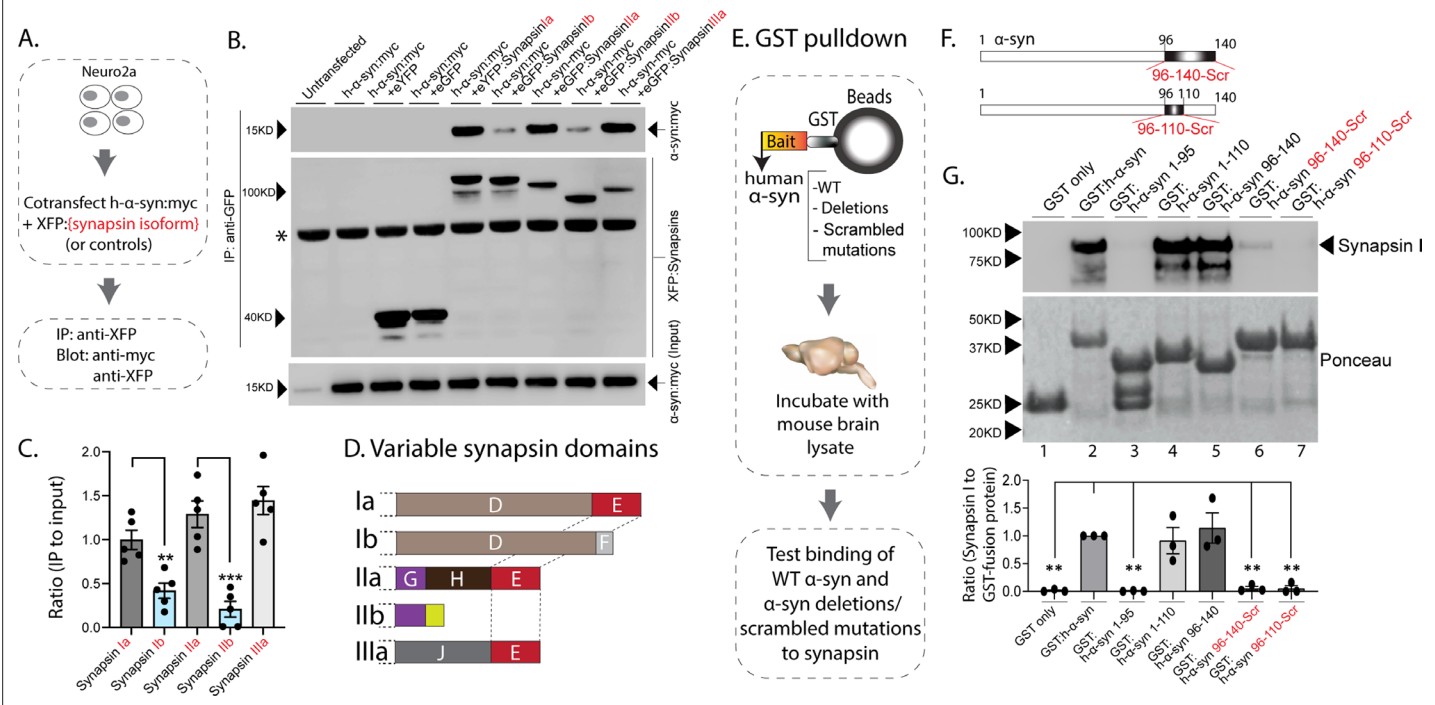

**Figure 2.** Interaction of synapsin isoforms with h-α-syn. (**A**) Workflow for co-immunoprecipitation experiments in neuro2a cells. (**B**) Western blots from co-immunoprecipitation experiments show that the synapsin isoforms Ia, IIa, and IIIa associate more robustly with h-α-syn (top panel), when compared to synapsins Ib and IIb (a non-specific band is marked with an asterisk). (**C**) Quantification of blots in (**B**) n=5, all data presented as mean ± SEM (a vs. b isoform, **p=0.003, ***p=0.0003, Student's t-test). (**D**) Schematic showing synapsin isoforms and their variable domains. Note that the E-domain is common between synapsins Ia, IIa, and Iia. (**E**) Workflow for pulldown of GST-tagged h-α-syn WT/deletions/scrambled mutations after incubation with mouse brain lysates. Equivalent amounts of immobilized GST α-syn variants were used. (**F**) Schematic showing α-syn regions that were scrambled (amino acids between 96–140 and 96–110). (**G**) Top: Samples from GST-pulldown were analyzed by NuPAGE and immunoblotted with an antibody against synapsin I (top panel). Bottom: Ponceau staining shows equivalent loading of fusion proteins. Note that full-length h-α-syn bound synapsin I from mouse brains (lane 2), while deletion of the h-α-syn C-terminus (amino acids 96–140, lane 3) eliminated this interaction. Lanes 4–7 show that the sequence within amino acids 96–110 of h-α-syn is critical for binding to synapsin I. All western blots are quantified below (n=3). Data presented as mean ± SEM (**p=0.003, **p=0.002, ns p=0.99, ns p=0.98, **p=0.004, **p=0.004, comparing to full-length h-α-syn, one-way ANOVA with Tukey's posthoc test).

The online version of this article includes the following source data for figure 2:

**Source data 1.** Tabular data and statistical analyses for graphs shown in panels C and G.

**Source data 2.** Full western blots for segments shown in panel B.

**Source data 3.** Full western blots for segments shown in panel G.

To test if the E-domain was *necessary* for enabling α-syn functionality, we generated a synapsin-Ia construct where the amino acid sequences of the E-domain were scrambled (*Figure 3A*, synapsin-Ia^ScrE). As shown previously, expression of WT synapsin-Ia enables α-syn-mediated synaptic attenuation in neurons lacking all synapsins (*Figure 1F*, leftmost panel). We reasoned that if the E-domain enabled α-syn functions and mediated synapsin/α-syn interactions in these experiments, scrambling this region should abolish such synapsin-dependent functions. Towards this, we used pHluorin assays in synapsin TKO neurons, asking if synapsin-Ia^ScrE would fail to reinstate α-syn functionality (schematic in *Figure 3B*). Indeed, while overexpressed h-α-syn was able to attenuate synaptic responses in the presence of WT-Synapsin-Ia in synapsin TKO neurons, Synapsin-Ia^ScrE failed to have any effect (*Figure 3C*), despite the detection of similar quantities of both at synapses (*Figure 3—figure supplement 1*). Analogously, in neuro2a co-immunoprecipitation experiments to test binding of WT-Synapsin-Ia or Synapsin-Ia^ScrE to α-syn, WT h-α-syn bound to Synapsin-Ia, but not to Synapsin-Ia^ScrE (*Figure 3D*), indicating that the E-domain is critical in mediating this interaction.

Next, we tested if the synapsin-E domain was *sufficient* for enabling α-syn functionality. Toward this, we first over-expressed the E-domain in synapsin TKO neurons, along with h-α-syn and sypHy (*Figure 3—figure supplement 2A*), with the overall intention of evaluating SV-recycling in this setting.

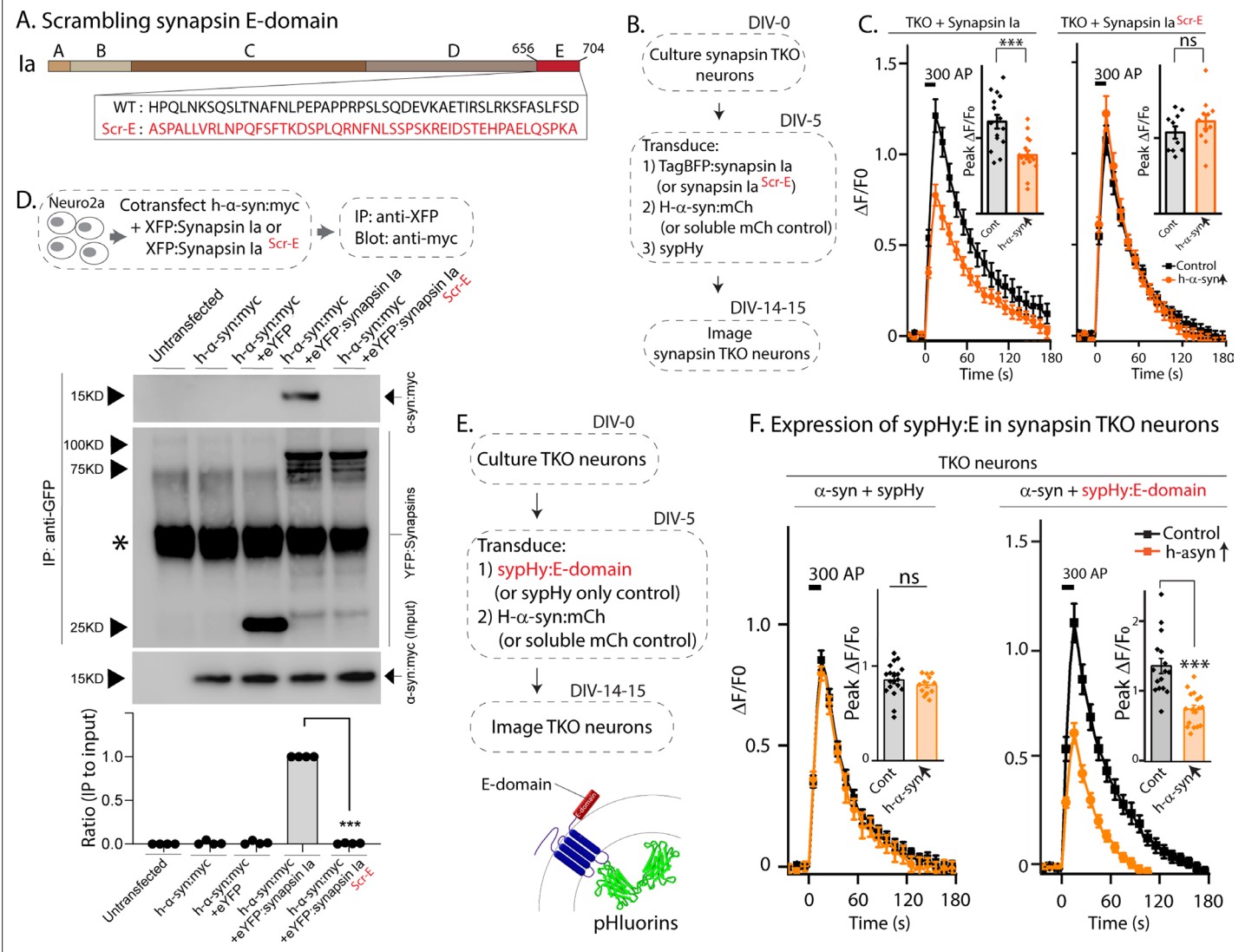

**Figure 3.** The synapsin E-domain is necessary and sufficient for enabling α-syn functionality. (**A**) Schematic showing synapsin Ia scrambled E-domain sequence (synapsin Ia[scr-E]). Numbers depict amino acid positions, letters in the inset depict amino-acids. Note that the WT amino acids are randomized in the scrambled mutant. (**B**) Design of sypHy experiments co-expressing synapsin Ia[scr-E] and h-α-syn in cultured neurons from synapsin TKO mice. (**C**) Stimulation-induced sypHy fluorescence traces (300 action potentials at 20 Hz, delivered at t=0 sec). Note that while h-α-syn attenuated sypHy fluorescence in synapsin TKO neurons expressing synapsin Ia, h-α-syn had no effect in neurons expressing synapsin Ia[scr-E]. Insets: Quantification of peak $\Delta F/F_0$ sypHy values (bars: mean ± SEM). Ten to 16 coverslips from at least three separate cultures were analyzed for each condition (***p=0.0007, ns p=0.67, one-way ANOVA with Tukey's posthoc analysis). (**D**) Top: Schematic for co-immunoprecipitation experiments, to test the interaction of h-α-syn with WT synapsin Ia or synapsin Ia[scr-E]. Neuro2a cells were co-transfected with myc-tagged α-syn and respective YFP-tagged synapsin Ia, and the YFP was immunoprecipitated. Bottom: Note that h-α-syn co-immunoprecipitated with synapsin Ia, but not synapsin Ia[scr-E]; quantification of the gels below (n=4, all data are means ± SEM ***p<0.001, Student's t test – a non-specific band is marked with an asterisk). (**E**) Schematic of experiments to test if the synapsin E-domain is sufficient to enable α-syn functionality in synapsin TKO neurons. Synapsin-E (a 46 amino acid sequence) was fused to the C-terminus of sypHy, so that upon expression in neurons, the E-domain would be present on the cytosolic surface of Svs. (**F**) SypHy fluorescence traces (mean ± SEM). Note that while h-α-syn (orange) was unable to attenuate SV recycling in synapsin TKO neurons (as expected), diminished synaptic responses were seen when the E-domain was present. Insets: Quantification of peak $\Delta F/F_0$ sypHy values (bars: mean ± SEM). Twelve 19 coverslips from at least three separate cultures were analyzed for each condition (ns p=0.89, ***p=2.8e-7, one-way ANOVA with Tukey's posthoc analysis).

The online version of this article includes the following source data and figure supplement(s) for figure 3:

**Source data 1.** Tabular data and statistical analyses for graphs shown in panels C, D and F.

**Source data 2.** Full western blots for segments shown in panel D.

**Figure supplement 1.** Similar synaptic of localization of synapsin Ia and synapsin Ia[scr-E] in synapsin TKO neurons.

**Figure supplement 1—source data 1.** Tabular data and statistical analyses for graph shown in panel C.

*Figure 3 continued on next page*

*Figure 3 continued*

**Figure supplement 1—source data 2.** Raw images.

**Figure supplement 2.** Synaptic targeting of synapsin E-domain constructs in synapsin null neurons.

**Figure supplement 2—source data 1.** Tabular data and statistical analyses for graphs shown in panels C and E.

**Figure supplement 2—source data 2.** Raw images.

However, we found that the E-domain by itself was not targeted to synapses (*Figure 3—figure supplement 2B*) – consistent with the known biology of synapsins (*Gitler et al., 2004*) – and expectedly, the E-domain had no effect on SV-recycling in pHluorin assays (*Figure 3—figure supplement 2C*). To allow the E-domain to operate in a context where it would be 'functionally available', we fused the synapsin E-domain to the C-terminus of sypHy. Since in this scenario, the small synapsin fragment would be localized to the cytosolic surface of SVs and target to synapses (*Figure 3E* and *Figure 3—figure supplement 2D*), we reasoned that such placing of the E-domain in the right cellular context may be sufficient to enable α-syn functionality. Indeed, forced targeting of the synapsin E-domain to the surface of SVs enhanced α-syn enrichment in synapses (*Figure 3—figure supplement 2E*), and restored α-syn mediated synaptic attenuation in synapsin null neurons (*Figure 3F*), suggesting that the E-domain was sufficient to reinstate the functional interplay between α-syn and synapsins. Collectively, the evidence makes a strong case that the synapsin E-domain is both necessary and sufficient to allow α-syn functionality at synapses.

Previous studies have shown that loss of all synapsins disrupt the tight clustering of SVs that is normally seen in cultured hippocampal neurons, leading to a reduced number of SVs within the bouton-boundary and an increase in vesicles spilling out into the adjacent axon [(*Orenbuch et al., 2012*), and see *Figure 4A*]. One possibility in our α-syn over-expression experiments is that excessive α-syn can bind to endogenous synapsin molecules (presumably via the E-domain) and prevent the normal functionality of synapsins (i.e. ability to cluster SVs). Dispersion of SVs can be quantified using 'full-width half-max' (FWHM) analysis, which is a quantitative measure of the extent of protein-dispersion at synapses (*Orenbuch et al., 2012*; *Wang et al., 2014*). Briefly, combined attenuation and dispersion of synaptic proteins would cause an increase in FWHM (see *Figure 4B*). As shown in *Figure 4C*, loss of synapsins lead to an overall reduction in the intensity of SV-staining at boutons (*Figure 4C*, left), as well as increased FWHM (*Figure 4C*, right). To examine SV dispersion in a α-syn over-expression setting, we cultured neurons from WT or synapsin TKO mice, and transduced either h-α-syn alone (in neurons from WT mice), or h-α-syn, along with various synapsin isoforms (in neurons from synapsin TKO mice – see strategy in *Figure 4D*). As shown in *Figure 4E*, over-expression of h-α-syn led to an attenuation/dispersion of SV-intensities (increased FWHM) in WT neurons, but had no effect in synapsin TKO neurons. Over-expression of Ia/IIa synapsin isoforms (but not Ib/IIb isoforms) also led to SV dispersion (IIIa was not tested). At first glance these data seem to contradict studies from many groups showing that α-syn clusters SVs (*Diao et al., 2013*; *Wang et al., 2014*; *Sun et al., 2019*), but we surmise that the AAV-mediated over-expression of α-syn in this setting creates a scenario where excessive α-syn binds to and displaces native synapsin molecules from SVs, or may disrupt synapsin-based protein condensates (*Hoffmann et al., 2021*; *Song and Augustine, 2023*; *Hoffmann et al., 2023*).

## Discussion

Precise organization of vesicles at synapses is critical for synaptic function (*Denker and Rizzoli, 2010*). Typically, each synapse has clusters of dozens to hundreds of SVs, and these vesicles are classified into different pools based on their ability to participate in exocytosis, and their physical proximity to the site of exocytosis (active zone). SVs within the readily releasable pool are docked at the active zone and can rapidly fuse with the plasma membrane in response to an action potential. On the other hand, SVs within the much larger reserve pool are distal to the active zone and are thought to help replenish SVs following exocytosis. Actively recycling vesicles comprise the recycling pool. Studies over several decades have shown that functional perturbation of synapsins selectively reduces the number of SVs in the reserve pool, establishing a role for synapsin in maintaining SV clusters within this pool [reviewed in *Zhang and Augustine, 2021*]. Previous studies have also explored the role of

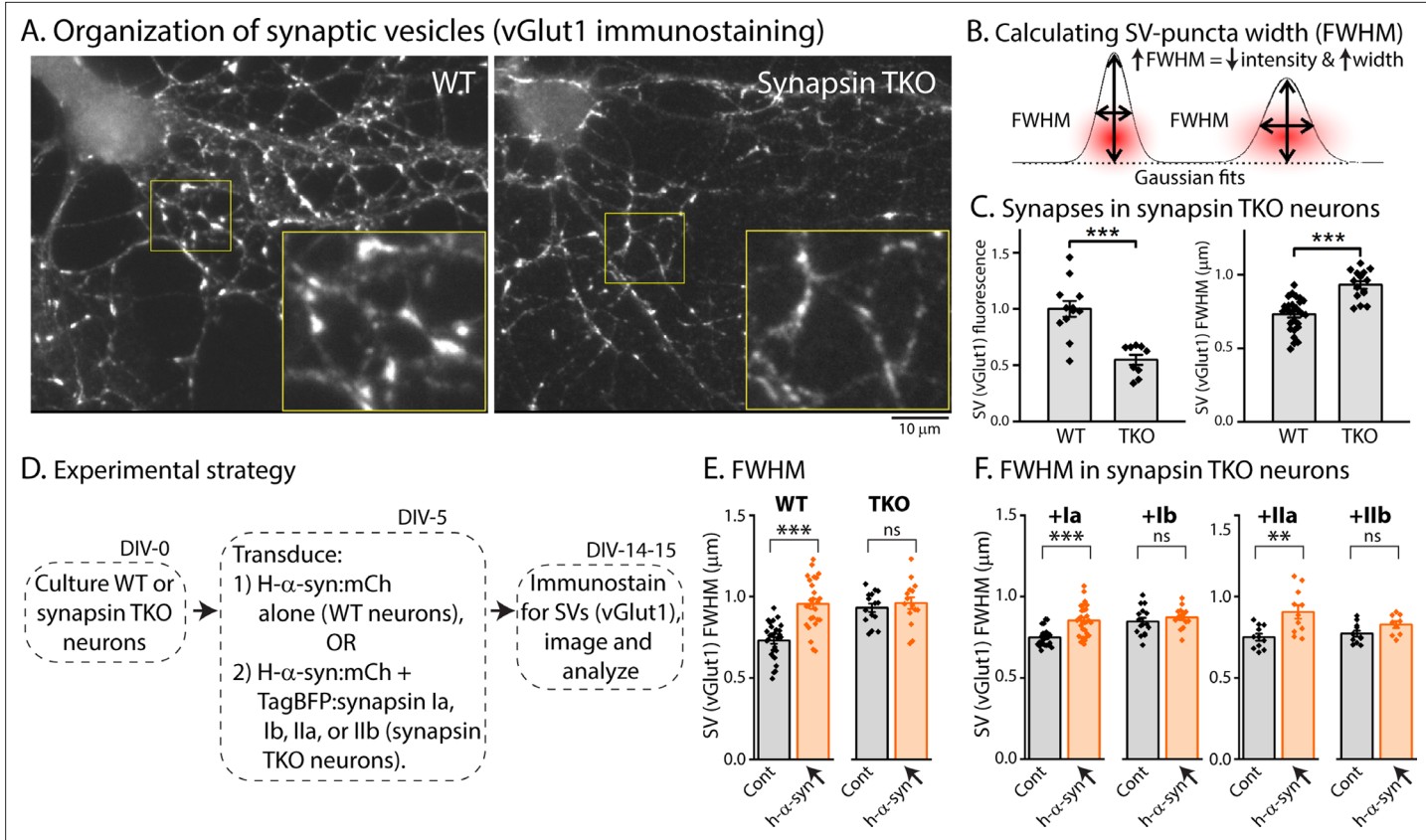

**Figure 4.** Synapsin-dependent redistribution of synaptic vesicles by α-syn overexpression. (**A**) Representative images from WT or synapsin TKO neurons immunostained with an SV marker (vGlut1); zoomed insets marked by yellow boundaries. Note that the compact clustering of SVs is lost in synapsin-null neurons. (**B**) FWHM as a quantitative means to determine spreading of fluorophores at synapses (also see Results). Note that an increase in FWHM corresponds to a decrease in intensity and increased spreading of fluorescence within a bouton. (**C**) Quantification of synaptic fluorescence in WT and synapsin TKO neurons. Overall intensities are decreased in TKO synapses (left), and FWHM is increased (right), compared to WT synapses; consistent with a spreading of SVs in the synapsin null setting. (**D**) Experimental plan to determine effects of h-α-syn over-expression on the overall distribution of SVs in WT and synapsin TKO neurons. (**E**) FWHM of vGlut1 staining at synapses is augmented by h-α-syn over-expression in WT neurons, but not in neurons from synapsin TKO mice. Reintroduction of synapsins Ia/IIa (but not Ib/IIb) in the setting of h-α-syn over-expression rescues the changes in vGlut1-FWHM (**F**). All data in this figure are represented as mean +/-SEM. Nine to 28 coverslips from at least three independent cultures were analyzed for C, E, and F (C, left: ***p=0.0006, Mann-Whitney U-test; right: see E; E: ***p=4e-8, ns p=0.92, one-way ANOVA with Tukey's posthoc analysis; F, left: ***p=2.7e-4, ns p=1.0, Kruskal-Wallis ANOVA with Dunn's posthoc test; F, right: **p=0.001, ns p=0.52, one-way ANOVA with Tukey's posthoc analysis).

The online version of this article includes the following source data for figure 4:

**Source data 1.** Raw images.

**Source data 2.** Tabular data and statistical analyses for graphs shown in panels C, E, and F.

the E-domain in various model systems. Microinjecting domain-E antibodies into lamprey giant axons dispersed the distal cluster of SVs (***Pieribone et al., 1995***), suggesting that this domain has a role in organizing the reserve pool. Injection of a peptide from the E-domain into squid giant synapses also dispersed the distal SV cluster, while docked SVs remained intact (***Hilfiker et al., 1998***), indicating that interfering with this domain in different ways resulted in the same phenotype – disruption of the reserve pool of SVs.

Thus, the current view is that the synapsin E-domain has an important role in maintaining the distal reserve pool SV clusters, although this domain has other independent roles in SV exocytosis that are not well defined (***Song and Augustine, 2015***). In this context, α-syn has also been long thought to play roles in SV organization and trafficking. First, the N-terminus of α-syn adopts a helical structure in the presence of small synaptic-like vesicles (***Burré et al., 2018***), and can also directly modulate vesicle shape (***Varkey et al., 2010***). In cell-free systems, recombinant α-syn can cluster synaptic-like vesicles (***Diao et al., 2013***; ***Sun et al., 2019***), and experiments with cultured neurons also support the idea that

α-syn can cluster SVs (*Wang et al., 2014*). For example, induced multimerization of α-syn at synapses clusters synaptic vesicles (*Wang et al., 2014*), and α-syn overexpression also diminished vesicle trafficking between synaptic boutons (*Scott and Roy, 2012*), which may reflect clustering of SVs by α-syn. Adjacent vesicles may also be directly tethered by α-syn (*Fusco et al., 2016*; *Lautenschläger et al., 2018*), thus α-syn-dependent organization and corralling of SVs are important clues to its function. Interestingly, a recent study showed that injection of an antibody to the N-terminus of α-syn into lamprey giant axons also led to a loss of SVs (*Fouke et al., 2021*) – resembling the SV disruption caused by synapsin E-domain injections (*Pieribone et al., 1995*; *Hilfiker et al., 1998*) – although both reserve and readily-releasable pools were depleted with α-syn injections. Our results support co-regulation of SV organization by both α-syn and synapsin, involving the synapsin E-domain. Additionally, α-syn has also been implicated in promoting SNARE complex formation (*Burré et al., 2010*), facilitating endocytosis (*Vargas et al., 2014*), and may participate in fusion-pore opening (*Logan et al., 2017*). Further work is needed to clarify whether these different effects of α-syn are linked, or whether they reflect functionality in distinct neuronal states (for instance in resting versus active neurons). In summary, our studies open the door to further mechanistic investigations into the functional interacting partners of α-syn, which will be important to uncover the myriad functions of this enigmatic protein. More broadly, our structure-function experiments place α-syn in a functional context with its interacting partners at the synapse, offering new insight into α-syn biology.

## Methods

### Animals, cell lines, antibodies, and DNA constructs

Animal studies were performed following the guidelines of the Ben-Gurion University Institutional Committee for Ethical Care and Use of Animals in Research (protocol IL-52-07-2019A) or of IACUC (UCSD protocol S19073). Synapsin triple knock-out (TKO) mice (RRID:MMRRC_041434-JAX) were backcrossed onto the C57BL/6 background as described previously (*Gitler et al., 2004*; *Boido et al., 2010*; *Shulman et al., 2015*), and C57BL/6JRccHsd mice (RRID:IMSR_ENV:HSD-043) served as WT controls. The following cell lines were obtained from ATCC and maintained using standard protocols: HEK293-T (RRID:CVCL_0063) and Neuro2A (TKG Cat#TKG 0509, RRID:CVCL_0470). Mycoplasma contamination was tested regularly. The following antibodies were used for immunofluorescence experiments: goat anti-vGlut1 (Synaptic systems Cat#135307, 1:1000), mouse anti-synapsin I (Synaptic systems Cat#106011, 1:1000), donkey anti-goat IgG NL-637 (R&D Systems Cat#NL002, 1:1000), donkey anti-mouse IgG NL-493 (R&D Systems Cat#NL009, 1:1000), VAMP2 (Synaptic systems Cat#104211, 1:1000). The following antibodies were used for biochemistry experiments: synapsin-1 (Abcam, Cat#ab254349), c-myc (Sigma, Cat#M4439, 1:500), GFP (Abcam, Cat#ab290, 1:5000). cDNAs of tagged synapsin isoforms, E domain variants, and fluorescent sensors [TagBFP:Synapsin-Ia/Ib/IIa/IIb/IIIa (*Gitler et al., 2004*), TagBFP:Synapsin-Ia$^{ScrE}$, TagBFP:E-domain, EGFP:E-domain, h-α-syn:m-Cherry, synaptophysinI-2XpHluorin (sypHy) and sypHy:E-domain] were obtained by PCR or digestion of existing plasmids and subcloned into an adeno-associated virus (AAV) backbone that contains the human synapsin promoter, the woodchuck post-transcriptional regulatory element (WPRE) and the bovine growth hormone polyadenlynation signal (bGHpA) (*Kügler et al., 2003*). XFP-tagged synapsin and the E-domain were previously described (*Gitler et al., 2004*). The sequence of the synapsin Ia E-domain was scrambled using the online tool Peptide Nexus (https://peptidenexus.com/article/sequence-scrambler). A synthetic DNA block (IDT) coding for the scrambled E domain was subcloned using Gibson-cloning (NEB). GST-α-syn 96–140 Scr and GST-α-syn 96–110 Scr plasmids were synthesized by GenScript (Piscataway, NJ, USA). All constructs were verified by sequencing.

### Hippocampal Cultures, AAV production, and transduction

Primary hippocampal cultures were obtained using standard procedures as described previously (*Tevet and Gitler, 2016*; *Stavsky et al., 2021*). In brief, P0-P2 pups of either sex were decapitated, and the brains were quickly removed. Dissected hippocampi were kept on ice in Hank's Balanced Salt Solution (HBSS, Biological Industries) supplemented with 20 mM HEPES at pH 7.4. Hippocampus pieces were incubated for 20 minutes at room temperature (RT) in a digestion solution consisting of HBSS, 1.5 mM CaCl$_2$, 0.5 mM EDTA, and 100 units of papain (Worthington, Cat#3127) activated with cysteine (Sigma, Cat#C7352). The brain fragments were then triturated gently two times using fire-polished

glass pipettes of decreasing diameter. Cells were seeded at a density of 80,000–100,000 cells per well on glass coverslips (Bar Naor, Cat#BN1001-12-1-CN) coated with poly-D-Lysine (Sigma, Cat#P0899). Cells were plated in Neurobasal-A medium (Thermo Fisher Scientific, Cat#10888022) supplemented with 2% B27 (Thermo Fisher Scientific, Cat#17504044), 2 mM Glutamax I (Thermo Fisher Scientific, Cat#35050038), 5% FBS (Biological Industries, Cat#04-007-1A), and 1 µg/ml gentamicin (Biological Industries, Cat#03-035-1C). After 24 hr, the medium was replaced with serum-free medium containing Neurobasal-A, 2 mM Glutamax I, and 2% B27. Cultures were maintained at 37 °C in a 5% $CO_2$ humidified incubator until used. For AAV production, HEK293-T cells were co-transfected with the targeting plasmid and two helper plasmids (pD1 and pD2). Hybrid AAV1/2 viral particles were produced as described previously (*Tevet and Gitler, 2016*). Neurons were transduced at 5–6 DIV by adding the viral particles to the growth medium and incubated for at least 7 days before imaging. Viral titers were individually adjusted to produce ~90% transduction efficiency. Expressed proteins were verified by western blot and immuno-labeling analysis.

## pHluorin assays, analysis, and fluorescence microscopy

### Vesicle recycling measurements

Neurons expressing sypHy were imaged at 12–14 DIV. Experiments were conducted in standard extracellular solution containing (in mM): NaCl 150, KCl 3, Glucose 20, HEPES 10, $CaCl_2$ 2, $MgCl_2$ 3, pH adjusted to 7.35. To block recurrent network activity, experiments were conducted in the presence of 10 µM DNQX [6,7-Dinitroquinoxaline-2,3 (1H,4H-dione)] (Sigma, Cat#D0540) and 50 µM APV [DL-2-Amino-5-phosphonopentanoic acid] (Sigma, Cat#A5282). After each experiment, the bath was perfused with saline in which 50 mM NaCl was replaced with $NH_4Cl$ to visualize the total vesicle population. For imaging, cultured neurons were placed in a stimulation chamber between parallel platinum wires (RC-49MFSH, Warner Instruments). Stimulation (300 bipolar pulses of 10 V/cm, each of a duration of 1 µs, at 20 Hz for 15 s), was delivered using a high-power stimulus-isolation unit (SIU-102B, Warner Instruments) driven by an isolated pulse-stimulator (2100, A-M Systems). Fifty images were obtained (43 at 0.2 Hz and then 7 images at 0.125 Hz) per experiment. At least 30 synaptic regions of interest (ROIs) were analyzed in each case. The baseline sypHy fluorescence ($F_0$) in each synapse was the average value measured in 6 pre-stimulation images. The fluorescence increment at time t [$\Delta F(t)=F(t)-F_0$] was normalized by the baseline value for each synapse. Synaptic $\Delta F(t)/F_0$ values were averaged across manually marked equal-size synaptic ROIs in each experiment (shown as symbols in bar-chart graphs). These were then averaged to obtain mean values for each experimental condition. Non-responding synaptic puncta were excluded. Experiments were performed using at least three independent cultures on different days. Fluorescent-tagged proteins were imaged before each experiment to confirm the presence of h-α-syn-mCherry and tagBFP-synapsins. All pHluorin assays (sypHy) were performed at room temperature. Fluorescence measurements were performed on a Nikon TiE inverted microscope driven by the NIS-elements software package (version 5.21.03, Nikon) (RRID:SCR_014329) https://www.nikoninstruments.com/Products/Software. The microscope was equipped with an Andor Neo 5.5 sCMOS camera (Oxford Instruments), a 40X0.75 NA Plan Fluor objective (Nikon, Cat#MRH00401), a 60X1.4 NA Apochromat oil immersion objective (Nikon, Cat#MRD01602), EGFP (Chroma Technology Corporation, Cat#49002) and Cy3 filter cubes (Chroma Technology Corporation, Cat#49004), BFP (Semrock Cat#LF405-A-000), mCherry (Semrock, Cat#TxRed-4040C) and Cy5 filter cubes (Semrock, Cat#CY5-404A), and a perfect-focus mechanism (Nikon).

### Quantification of endocytosis rates

Endocytosis rates were assessed based on the decay of sypHy fluorescence after the termination of stimulation. Data were fit with a single-exponential decay-function (32 data points, 160 s) starting 5 s after stimulation cessation. The function is:

$$y = y_0 + Ae^{\frac{-t}{\tau}} , \qquad (1)$$

where $A$ is an amplitude, $y_0$ is an offset and $\tau$ is the time constant, assuming stimulation starts at t=0 for all traces.

Fit results were discarded if $\tau$ was longer than 160 s (the duration of the data being fit).

## Measurement of the recycling pool relative size

The relative size of the recycling pool was calculated based on imaging of cumulative exocytosis. Cumulative exocytosis was achieved by blocking SV reacidification by adding 1 μM bafilomycin A1 (Enzo Life Sciences, Cat#BML-CM110-0100) to the bathing medium itemized above. Neurons were imaged at 0.2 Hz throughout the experiment. Six baseline images were acquired, and stimulation was applied at t=0 for 2 min at 20 Hz (2400 action potentials), until saturation. The fluorescence of the total vesicle population ($F_{max}$) was measured at the end of each experiment by perfusing the chamber with $NH_4Cl$-saline. Synaptic sypHy signals were measured from at least 30 ROIs as explained above, subtracting from each its mean baseline value and normalizing it by $F_{max}$. The relative size of the recycling pool was defined as the ratio of the mean of the last three data points (at saturation, before $NH_4Cl$ exposure) and $F_{max}$.

## Evaluation of width of SV distribution

Neurons were fixed using 4% paraformaldehyde diluted from a 16% stock (Electron Microscopy Sciences, Cat#15710) in phosphate-buffered saline (Biological Industries, Cat#02-020-1A) for 10 min, washed thoroughly with PBS and permeabilized with PBS supplemented with 0.1% triton X100 (Sigma, Cat#X100-500ML) for 1 min and washed three times. Blocking solution (PBS with 5% skim milk powder; Sigma, Cat#70166–500 G) was applied for 1 hr. The coverslips were incubated for 1 hr with the indicated primary antibodies (see above) in blocking solution at RT, washed X3, and then incubated with secondary antibodies in blocking solution for 1 hr at RT. Finally, the preps were washed X3 and mounted using immumount (Thermo Fisher Scientific, Cat#9990402). Neurons were imaged using a 60X1.4 NA oil-immersion Apochromat objective (Nikon, Cat#MRD01602). Linear profiles were drawn manually along axonal segments and through synaptic puncta in the vGlut1 channel using NIS elements (Nikon). The intensity profiles were imported into Origin (2023) (RRID:SCR_014212) http://www.originlab.com/index.aspx?go=PRODUCTS/Origin and fit individually with Gaussian functions. The standard deviation parameter (σ) of the fit was extracted, and the FWHM was calculated thus:

$$FWHM = 2\sqrt{ln(4)}\sigma = 2.355\sigma \qquad (2)$$

Average FWHM values were calculated per experiment.

## Semi quantitative determination of synaptic fluorescence intensity

Synaptic puncta were detected as already described (*Orenbuch et al., 2012*), using an in-house thresholding algorithm in which the threshold is iteratively decreased, detected objects are filtered based on their area and roundness (>0.7), saved, and then blanked to not be chosen again. Subsequently, objects that the user judges by eye not to represent synaptic puncta, or those which are out of focus are removed manually. The peak fluorescence at the center-of-mass (2x2 pixels in size) in each punctum was recorded, and synaptic intensity values were averaged per image. All experimental conditions of fluorescence intensity experiments were performed and processed; in each imaging session, the mean intensity value of the control condition was used to normalize all recorded values to reduce inter-session variability. Normalized intensity values were then averaged across sessions. Experiments were performed in at least three independent cultures.

## Measurement of synaptic enrichment

Synaptic enrichment was measured as described previously (*Atias et al., 2019*). Neurons were transduced at 5 DIV with either sypHy or sypHy-E-domain, h-α-syn-mCherry and soluble tagBFP as a measure of local volume. At 14 DIV, the neurons were fixed and immunostained with anti-vGlut1 antisera to visualize synaptic puncta. Analysis lines (at least 30) were drawn in each image, starting in the axon, through a synapse, and into the surrounding background. The intensity profiles corresponding to the h-α-syn-mCherry and tagBFP channels were fit with a Gaussian function to determine the axonal ($F_{axon}$) and synaptic ($F_{syn}$) intensity values of each color thus:

$$F = F_{axon} + F_{syn}e^{\frac{(x-x_c)^2}{2w^2}}, \qquad (3)$$

where $x_c$ is the center of the Gaussian (the synaptic center) and $w$ is its width.

The percentage of synaptic enrichment (E%) of h-α-syn-mCherry is defined thus:

$$E\% = \left( \frac{F_{syn}(red)/F_{axon}(red)}{F_{syn}(blue)/F_{axon}(blue)} - 1 \right) * 100 \tag{4}$$

Protocol available at https://doi.org/10.17504/protocols.io.bp2l6xyx5lqe/v1.

## Biochemical assays and evaluation

### Preparation of brain and neuro2A lysates

Whole mouse brains were homogenized with a Dounce tissue grinder in neuronal protein extraction reagent (N-PER) (Thermo Scientific, Cat#87792) containing protease/phosphatase inhibitors (Cell Signaling, Cat#5872). Triton X-100 (Sigma, Cat#X100-500ML) was added to a final concentration of 1%, and the samples were incubated with rotation for 1 hr at 4 °C. Samples were centrifuged at 10,000×$g$ for 10 min at 4 °C, and the supernatant was collected. To obtain Neuro2A lysates, cells were washed with 1 X PBS three times and incubated 5 min on ice in the presence of N-PER reagent supplemented with protease inhibitors. Samples were centrifuged at 10,000×$g$ for 10 min at 4 °C to remove cellular debris. After obtaining the brain and Neuro2A lysates, we measured protein concentration (DC Protein Assay Kit II, Biorad), and samples were used in subsequent experiments. Protocol available at https://doi.org/10.17504/protocols.io.5jyl8pey7g2w/v1.

### Immunoprecipitations and western blots analysis

Immunoprecipitations were performed using 1–2 mg of total protein. Samples were incubated overnight with the indicated antibody at 4 ° C, followed by the addition of 50 µl of protein G-agarose beads (Thermo Fisher Scientific, Cat#20397). Immunoprecipitated proteins were recovered by centrifugation at 2500×rpm for 2 min, washed three times with a buffer containing PBS and 0.15% Triton X-100 (Sigma, Cat#X100-500ML). The resulting pellets were resuspended in 20 µl of 1 X NuPAGE LDS sample buffer (Thermo Fisher Scientific Cat#NP007) and incubated at 95 °C for 10 min. Samples were separated by NuPAGE 4 to 12% Bis-Tris polyacrylamide gels (Thermo Fisher Scientific, Cat#NP0335BOX), and transferred to a 0.2 µM PVDF membrane (Thermo Fisher Scientific, Cat#LC2002), using the Mini Blot Module system (Thermo Fisher Scientific). PVDF membranes were first fixed with 0.2% PFA 1 x PBS per 30 min at room temperature. Then, membranes were washed three times for 10 min in PBS with 0.1% Tween 20 Detergent (TBST) and blocked for 1 hr in TBST buffer containing 5% dry milk, and then incubated with the indicated primary antibody for 1 hr in blocking solution, washed three times for 10 min each and incubated with HRP-conjugated secondary antibodies (RRID:AB_2819160, RRID:AB_2755049). After antibody incubations, membranes were again washed three times with TTBS buffer, and protein bands were visualized using the ChemiDoc Imaging System (Bio-Rad) and quantified with Image Lab software version 6.1 from Bio-Rad (RRID:SCR_014210) http://www.bio-rad.com/en-us/sku/1709690-image-lab-software. Protocol available at https://doi.org/10.17504/protocols.io.36wgq3ep5lk5/v1.

## GST fusion proteins production

Full-length recombinant human WT α-syn (Addgene #213498), α-syn 1–95 (Addgene #213499), α-syn 1–110 (Addgene #213500), α-syn 96–140 (Addgene #213501), α-syn 96–140 Scr (Addgene #213502) and α-syn 96–110 Scr (Addgene #213503) were expressed in *Escherichia coli BL21 (DE3)* (New England Biolab, Cat#C2530H) using the bacterial expression vector pGEX-KG myc (Addgene #209891). Following transformation, protein expression was induced with 0.05 mM IPTG (isopropyl-β-d-thiogalactopyranoside), and either incubated at 37 °C for 2 hr or at room temperature for 6 h, with shaking. The cells grown on Terrific Broth (Thermo Scientific, Cat#BP9728-2) were harvested by centrifugation at 4500 × $g$ at 4 °C for 20 min, and pellets were stored at –80 °C until use. For protein purification, protein pellets were resuspended in 30 ml Lysis Buffer containing 1 X PBS, 0.5 mg/ml lysozyme, 1 mM PMSF, DNase, and EDTA-free protease cocktail inhibitor (Roche, Cat#11836170001) for 15 min on ice, briefly sonicated (3 sets with 33 strikes and 30 second breaks on ice between sets), and removed the insoluble material by centrifugation at 15,000 × $g$ at 4 °C for 30 min. The clarified lysate was incubated with 500 µl of glutathione-Sepharose 4B (Sigma, Cat#17-0756-01), preequilibrated with 1 X PBS containing 0.1% Tween 20 and 5% glycerol (binding buffer), on a tumbler at 4 °C overnight. The GST-bound proteins were washed four times with 30 ml binding buffer and maintained

at 4 °C for pull-down assays. Protocol available at https://doi.org/10.17504/protocols.io.4r3l22y14l1y/v1.

## Pull-down assays

To pull down Synapsin Ia from brain lysates, 1–2 mg of the sample was incubated with 25–50 µg of glutathione beads containing GST fusion proteins for 12–16 hr. The mixtures were washed three times with 1 X PBS with 0.15% Triton X-100 (Sigma, Cat#X100-500ML), and then resuspended in 20 µl of 1 X NuPAGE LDS sample buffer for NuPAGE and immunoblotted analysis. Protocol available at https://doi.org/10.17504/protocols.io.x54v9pw5pg3e/v1.

## Statistical analysis

Results are expressed as mean ± SEM values, and symbols are the results of individual experiments. The normality of the distribution was tested using the Shapiro-Wilk test. Pairs of datasets were compared using the two-sided students' t-test when deemed to be normally distributed; otherwise, Mann-Whitney's non-parametric u-test was used. Multiple comparisons of normally distributed datasets were performed using one-way ANOVA or two-way ANOVA, followed by Tukey's post-hoc analysis. When the distribution of one or more of the compared conditions was deemed not to be distributed normally, the Kruskal-Wallis test was used, with Dunn's test for posthoc analysis. Outliers were identified using Grubbs's test. Statistical significance was set at a confidence level of 0.05 for all tests. In all figures: 'ns' denotes $p \geq 0.05$; * $p<0.05$; ** $p<0.01$; and *** $p<0.001$. Statistical analysis was performed using Origin (2023) (RRID:SCR_014212) http://www.originlab.com/index.aspx?go=PRODUCTS/Origin or GraphPad Prism software (version 6) (RRID:SCR_002798) http://www.graphpad.com.

## Materials availability

New reagents are available from the corresponding authors upon request. New plasmids, as indicated in the Key Resources Table will be available through Addgene.

## Acknowledgements

This work was supported by grant 2019248 from the United States-Israel Binational Science Foundation to Daniel Gitler and Subhojit Roy, grants 1310/19 and 189/22 from the Israel Science Foundation, and the Bergida Endowment on Parkinson's Disease research to Daniel Gitler, grants to Subhojit Roy from the NINDS (R01NS111978), the Farmer Family Foundation, and a NINDS P30NS047101 grant to the UCSD microscopy core. This research was also funded in whole or in part by Aligning Science Across Parkinson's [ASAP-020495] through the Michael J Fox Foundation for Parkinson's Research (MJFF). For the purpose of open access, the author has applied a CC BY public copyright license to all Author Accepted Manuscripts arising from this submission (CC-BY 4.0). Leonardo Parra-Rivas was supported by a postdoctoral fellowship from the American Parkinson's Disease Association (APDA) and the Parkinson's Foundation Launch award PF-LAUNCH-1046253.

## Additional information

### Funding

| Funder | Grant reference number | Author |
| --- | --- | --- |
| United States-Israel Binational Science Foundation | 2019248 | Subhojit Roy<br>Daniel Gitler |
| Israel Science Foundation | 1310/19 | Daniel Gitler |
| Israel Science Foundation | 189/22 | Daniel Gitler |
| Bergida Endowment on Parkinson's Disease Research | | Daniel Gitler |

| Funder | Grant reference number | Author |
|---|---|---|
| National Institute of Neurological Disorders and Stroke | R01NS111978 | Subhojit Roy |
| National Institute of Neurological Disorders and Stroke | P30NS047101 | Subhojit Roy |
| Farmer Family Foundation | | Subhojit Roy |
| Aligning Science Across Parkinson's | ASAP-020495 | Subhojit Roy |
| American Parkinson Disease Association | | Leonardo A Parra-Rivas |
| Parkinson's Foundation | PF-LAUNCH-1046253 | Leonardo A Parra-Rivas |

The funders had no role in study design, data collection and interpretation, or the decision to submit the work for publication.

## Author contributions

Alexandra Stavsky, Conceptualization, Data curation, Formal analysis, Validation, Investigation, Visualization, Methodology; Leonardo A Parra-Rivas, Conceptualization, Data curation, Formal analysis, Validation, Investigation, Methodology; Shani Tal, Jen Riba, Data curation, Formal analysis, Validation, Investigation, Visualization; Kayalvizhi Madhivanan, Methodology; Subhojit Roy, Conceptualization, Data curation, Supervision, Funding acquisition, Visualization, Writing – original draft, Project administration, Writing – review and editing; Daniel Gitler, Conceptualization, Formal analysis, Supervision, Funding acquisition, Validation, Visualization, Methodology, Writing – original draft, Project administration, Writing – review and editing

## Author ORCIDs

Alexandra Stavsky ⓘ http://orcid.org/0000-0002-8209-3524
Leonardo A Parra-Rivas ⓘ http://orcid.org/0000-0002-6707-1255
Jen Riba ⓘ http://orcid.org/0009-0004-5016-5636
Subhojit Roy ⓘ https://orcid.org/0000-0002-1571-2735
Daniel Gitler ⓘ https://orcid.org/0000-0001-9544-3610

## Ethics

Animal experimentation protocols used in this study were approved by the Ben-Gurion University Institutional Committee for Ethical Care and Use of Animals in Research (protocol IL-52-07-2019A) or the UCSD Institutional Animal Care and Use Committee (IACUC protocol S19073). Protocols were executed in accordance with their respective guidelines.

Reviewer #1 (Public Review): https://doi.org/10.7554/eLife.89687.3.sa1
Author response https://doi.org/10.7554/eLife.89687.3.sa2

# Additional files

## Supplementary files
• MDAR checklist

## Data availability

Source data files containing the numerical data used to generate Figure 1 B, C, F, G, Figure 1—figure supplement 1 A, B, D, E, Figure 2 C, G, Figure 3 C, D, F, Figure 3—figure supplement 1 C, Figure 3—figure supplement 2 C, E, Figure 4 C, E, F are appended to the corresponding figure legends. Raw source data (western blots, images, imaging time sequences) were uploaded to Zenodo.

The following datasets were generated:

| Author(s) | Year | Dataset title | Dataset URL | Database and Identifier |
|---|---|---|---|---|
| Parra Rivas LA | 2023 | Synapsin E-domain is essential for α-synuclein function | https://doi.org/ 10.5281/zenodo. 10254061 | Zenodo, 10.5281/ zenodo.10254061 |
| Gitler D | 2024 | Synapsin E-domain is essential for α-synuclein function, Imaging data | https://doi.org/ 10.5281/zenodo. 11067289 | Zenodo, 10.5281/ zenodo.11067289 |

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

# Appendix 1

## Appendix 1—key resources table

| Reagent type (species) or resource | Designation | Source or reference | Identifiers | Additional information |
|---|---|---|---|---|
| Strain, strain background (*Mus musculus*, both sexes) | WT mice, C57BL/6JRccHsd | Inotiv (Envigo) | RRID:IMSR_ENV:HSD-043 | |
| Genetic reagent (*Mus musculus*, both sexes) | Synapsin TKO mice | *Gitler et al., 2004*; *Boido et al., 2010* | RRID:MMRRC_041434-JAX | Rederived on C57Bl/6 background |
| Cell line (Human) | HEK293-T | ATCC | RRID:CVCL_0063 | |
| Cell line (Mouse) | Neuro2A | ATCC | RRID:CVCL_0470 | TKG Cat#TKG 0509 |
| Antibody | anti-vGlut1 (goat polyclonal) | Synaptic Systems | 135 307 | (IF: 1:1000) |
| Antibody | anti-synapsin I (mouse monoclonal) | Synaptic Systems | 106 011 | (IF: 1:1000) |
| Antibody | anti-synapsin-1 (Recombinant) | Abcam | ab254349 | (WB: 1:1000) |
| Antibody | anti-c-myc (mouse monoclonal clone 9E10) | Sigma-Aldrich | M4439 | (WB 1:500) |
| Antibody | anti-GFP (Rabbit polyclonal) | Abcam | ab290 | (WB 1:5000) |
| Antibody | anti-VAMP2 (mouse monoclonal) | Synaptic Systems | 104 211 | (IF 1:1000) |
| Antibody | anti-goat IgG, NL-637 label (Donkey polyclonal) | R&D systems | NL002 | (IF 1:1000) |
| Antibody | anti-mouse IgG, NL-493 label (Donkey polyclonal) | R&D systems | NL009 | (IF 1:1000) |
| Antibody | anti-rabbit IgG H&L, HRP (Goat polyclonal) | Abcam | ab205718 | (WB 1:1000) |
| Antibody | anti-mouse IgG H&L, HRP (Goat polyclonal) | Abcam | ab205719 | (WB 1:1000) |
| Recombinant DNA reagent | pD1 | *Tevet and Gitler, 2016* | AAV1 | Cap1, Rep2, E2A, E4, VA |
| Recombinant DNA reagent | pD2 | *Tevet and Gitler, 2016* | AAV2 | Cap2, Rep2, E2A, E4, VA |
| Recombinant DNA reagent | pAAV2-hSyn-sypHy | *Orenbuch et al., 2012* | DG87 | hSyn (promoter) Synaptophysin I –2XpHluorin |
| Recombinant DNA reagent | pAAV2-hSyn-sypHy:E-domain | This paper | DG189 | hSyn (promoter) Synaptophysin I –2XpHluorin – Synapsin Ia E domain |
| Recombinant DNA reagent | pAAV2-hSyn-tagBFP | This paper | DG138 | hSyn (promoter) tagBFP |
| Recombinant DNA reagent | pAAV2-hSyn-tagBFP:E-domain | This paper | DG201 | hSyn (promoter) tagBFP-Synapsin Ia E domain |
| Recombinant DNA reagent | pAAV2-hSyn-tagBFP:ScrE-domain | This paper | DG200 | hSyn (promoter) tagBFP- Scrambled Synapsin Ia E domain |
| Recombinant DNA reagent | pAAV2-hSyn-EGFP:E-domain | This paper | DG187 | hSyn (promoter) EGFP-Synapsin Ia E domain |
| Recombinant DNA reagent | pAAV2-hSyn-h-α-syn-mCherry | *Atias et al., 2019* | DG79 | hSyn (promoter) human-alpha-synuclein-mCherry |
| Recombinant DNA reagent | pAAV2-hSyn- mCherry | *Atias et al., 2019* | DG97 | hSyn (promoter) mCherry |
| Recombinant DNA reagent | pEYFPC1-Synapsin Ia | *Gitler et al., 2004* | Syn03 | CMV (enhancer +promoter) EYFP-Synapsin Ia |
| Recombinant DNA reagent | pEGFPC1-Synapsin Ib | *Gitler et al., 2004* | Syn65 | CMV (enhancer +promoter) EGFP-Synapsin Ib |

*Appendix 1 Continued on next page*

*Appendix 1 Continued*

| Reagent type (species) or resource | Designation | Source or reference | Identifiers | Additional information |
|---|---|---|---|---|
| Recombinant DNA reagent | pEGFPC1-Synapsin IIa | *Gitler et al., 2004* | Syn50 | CMV (enhancer +promoter) EGFP-Synapsin IIa |
| Recombinant DNA reagent | pEGFPC2-Synapsin IIb | *Gitler et al., 2004* | Syn73 | CMV (enhancer +promoter) EGFP-Synapsin IIb |
| Recombinant DNA reagent | pEGFPC1-Synapsin IIIa | *Gitler et al., 2004* | Syn59 | CMV (enhancer +promoter) EGFP-Synapsin IIIa |
| Recombinant DNA reagent | pEYFPC1-Synapsin Ia ScrE | This paper | Syn88 | CMV (enhancer +promoter) EYFP-Synapsin Ia Scrambled E domain |
| Recombinant DNA reagent | pEYFPC1 | Clontech | | CMV (enhancer +promoter) EYFP |
| Recombinant DNA reagent | pEGFPC1 | Clontech | | CMV (enhancer +promoter) EGFP |
| Recombinant DNA reagent | h-α-syn-myc | This paper | pLP351 | EF-1 alpha promoter (pCCL backbone) |
| Recombinant DNA reagent | GST | *Parra-Rivas et al., 2023* | Addgene # 209891 | pGEX-KG myc |
| Recombinant DNA reagent | GST-h-α-syn FL | This paper | Addgene # 213498 | pGEX-KG myc backbone |
| Recombinant DNA reagent | GST-h-α-syn 1–95 | This paper | Addgene # 213499 | pGEX-KG myc backbone |
| Recombinant DNA reagent | GST-h-α-syn 1–110 | This paper | Addgene # 213500 | pGEX-KG myc backbone |
| Recombinant DNA reagent | GST-h-α-syn 96–140 | This paper | Addgene # 213501 | pGEX-KG myc backbone |
| Recombinant DNA reagent | GST-h-α-syn FL Scrambled 96–110 | This paper | Addgene # 213503 | pGEX-KG myc backbone |
| Recombinant DNA reagent | GST-h-α-syn FL Scrambled 96–140 | This paper | Addgene # 213502 | pGEX-KG myc backbone |
| Recombinant DNA reagent | pAAV2-hSyn-tagBFP-Synapsin Ia | This paper | DG199 | hSyn (promoter) tagBFP-Synapsin Ia |
| Recombinant DNA reagent | pAAV2-hSyn-tagBFP-Synapsin Ib | This paper | DG150 | hSyn (promoter) tagBFP-Synapsin Ib |
| Recombinant DNA reagent | pAAV2-CMV/CBAP-tagBFP-Synapsin IIa | *Orenbuch et al., 2012* | DG18 | CMV-CBAP (promoter) tagBFP-Synapsin IIa |
| Recombinant DNA reagent | pAAV2-hSyn-tagBFP-Synapsin IIb | This paper | DG160 | hSyn (promoter) tagBFP-Synapsin IIb |
| Recombinant DNA reagent | pAAV2-hSyn-tagBFP-Synapsin IIIa | This paper | DG153 | hSyn (promoter) tagBFP-Synapsin IIIa |
| Chemical compound, drug | Bafilomycin A1 | Enzo Life Sciences | BML-CM110-0100 | |
| Chemical compound, drug | 6,7-Dinitroquinoxaline-2,3 (1H,4H-dione), DNQX | Sigma-Aldrich | D0540 | |
| Chemical compound, drug | DL-2-Amino-5-phosphonopentanoic acid, APV | Sigma-Aldrich | A5282 | |
| Software, algorithm | Origin Pro 2023 | Originlab | RRID:SCR_014212 | |
| Software, algorithm | GraphPad Prism 6 | Graphpad | RRID:SCR_002798 | |
| Software, algorithm | NIS-elements AR 5.21.03 | Nikon | RRID:SCR_014329 | |
| Other | N-PER | Thermo Scientific | 87792 | Methods: Biochemical assays and evaluation |
| Other | protease/ phosphatase inhibitors | Cell Signaling | 5872 | Methods: Biochemical assays and evaluation |

*Appendix 1 Continued on next page*

*Appendix 1 Continued*

| Reagent type (species) or resource | Designation | Source or reference | Identifiers | Additional information |
|---|---|---|---|---|
| Other | Neurobasal-A medium | Thermo-Fisher Scientific | 10888022 | Methods: Hippocampal Cultures, AAV production, and transduction |
| Other | B27 supplement | Thermo-Fisher Scientific | 17504044 | Methods: Hippocampal Cultures, AAV production, and transduction |
| Other | Glutamax I | Thermo-Fisher Scientific | 35050038 | Methods: Hippocampal Cultures, AAV production, and transduction |
| Other | Fetal Bovine Serum, European Grade | Biological Industries | 04-007-1A | Methods: Hippocampal Cultures, AAV production, and transduction |
| Other | Gentamicin | Biological Industries | 03-035-1C | Methods: Hippocampal Cultures, AAV production, and transduction |
| Other | immumount | Thermo Scientific | 9990402 | Methods: pHluorin assays, analysis, and fluorescence microscopy |

