## [Editor Report · eLife assessment]

Alpha-synuclein is a synaptic vesicle associated protein that is linked to a number of neurodegenerative disorders. In this manuscript, the authors provide **compelling** evidence of alpha-synuclein's interaction with E-domain synapsins as the main culprit mediating the suppression of neurotransmitter release and synaptic vesicle recycling by alpha-synuclein. This **important** work provides molecular mechanisms underlying alpha-synuclein functions.

---

## [Referee Report · Reviewer #1 (Public Review)]

This is a short but important study. Basically, the authors show that α-synuclein overexpression's negative impact on synaptic vesicle recycling is mediated by its interaction with E-domain containing synapsins. This finding is highly relevant for synuclein function as well as for the pathophysiology of synucleinopathies. The data is clear, functional analysis is highly adequate.

---

## [Author Response]

The following is the authors’ response to the original reviews.

Reviewer #1This is a short but important study. Basically, the authors show that α-synuclein overexpression's negative impact on synaptic vesicle recycling is mediated by its interaction with E-domain containing synapsins. This finding is highly relevant for synuclein function as well as for the pathophysiology of synucleinopathies. While the data is clear, functional analysis is somewhat incomplete.(1) The authors should present a clearer dissociation of endocytosis and exocytosis under the various conditions they study. They should quantify the rate of rise and decay of pHluorin signals.

1. In addition, I strongly recommend a few additional experiments with and without a vATPase inhibitor such as bafilomycin to estimate the relative effects on exo- vs. endocytosis. As the authors are aware bafilomycin will mask the re-acidification /endocytosis component, thus revealing pure exocytosis and thus enabling quantification of endocytosis with minimal contamination from exocytosis.

In the revised version, we analyzed and quantified exocytosis and endocytosis separately, with bafilomycin experiments, as the reviewer suggested (new data, Fig. 1- Fig. Supp. 1A-B). Overexpression of human alpha-synuclein only attenuated exocytosis in neurons that also expressed synapsins (WT neurons and synapsin TKO neurons transduced with synapsin Ia). In parallel, we also examined endocytosis by calculating the time-constant of the decay in the fluorescence of sypHy during the endocytotic phase (Fig. 1- Fig. Supp. 1C-E). Previous studies have shown that after brief stimulus-trains – like those used in our study (20Hz/300AP) – most endocytosis occurs after the cessation of stimulation 1. Expression of human alpha-synuclein did not alter the endocytosis time-constant in any of our experiments. To summarize, the interaction of alpha-synuclein with the synapsin E domain was required for alpha-synuclein induced attenuation of exocytosis, but not endocytosis.

**Reviewer #2**
...The paper will be improved significantly if additional experiments are added to expand and provide a more mechanistic understanding of the effect of α-syn and the intricate interplay between synapsin, α-syn, and the SV. For an enthusiastic reader, the manuscript as it looks now with only 3 figures, ends prematurely. Some of the experiments above or others could complement, expand and strengthen the current manuscript, moving it from a short communication describing the phenomenon to a coherent textbook topic. Nevertheless, this work provides new and exciting evidence for the regulation of neurotransmitter release and its regulation by synapsin and α-syn.(1) Did the authors try to attach E-domain for example to synapsin Ib and restore α-syn inhibition with synapsin Ib-E?

This is an interesting idea, but in previous studies, we found that synapsin Ib does not associate with synaptic vesicles2, so it will not be present at the right location to be able to restore alpha-synuclein induced synaptic attenuation. We have also seen that this mis-localization alters synaptic properties (unpublished).

(2) Was the expression level of Synapsin-IaScrE examined and compared to WT Synapsin-Ia in Fig 3?

Yes, this data is now shown in Fig. 3-Fig. Supp. 1.

(3) Were SVs dispersed in α-syn overexpression as predicted?

We interpret the reviewer’s question and reasoning as follows. If alpha-synuclein binds to the E-domain of synapsin, a prediction in the alpha-synuclein over-expression scenario is that the overabundance of alpha-synuclein molecules would bind to and sequester the E-domain synapsins away from synaptic vesicles. In the absence of E-domain synapsins, the synaptic-vesicle clustering effects of synapsins would be lost, and there would be dispersion of synaptic vesicles. We tested this prediction, which is now shown in an additional figure (new data, Fig. 4). Indeed, the AAV-mediated over-expression of alpha-synuclein leads to a dispersion of synaptic vesicles, and this dispersion is dependent on synapsins Ia and Ib, but not IIa and IIb (please see Fig. 4D-E in the revised manuscript). Appropriate text is also added, starting with “Previous studies have shown that loss of all synapsins...” presents this data and interprets it.

(4) How does this study coincide with the effects of α-syn on fusion pore and endocytosis? This should be at least discussed. It is also possible that the effects of α-syn on endocytosis might affect the results as if endocytosis is affected, SVs number and distribution will be also affected.

It is difficult to reconcile our data with the idea that alpha-synuclein facilitates fusion-pore opening, as proposed by the Edwards lab 3. In fact, its difficult to reconcile this concept with their own previous data, showing that alpha-synuclein over-expression attenuates SV-recycling 4. As mentioned above, modulation of endocytosis does not seem to be a major factor in our experiments, though this does not rule out a physiologic role for alpha-synuclein in endocytosis, since all our experiments are based on over-expression paradigms. Future experiments looking at phenotypes after acute alpha-synuclein knockdown may provide more clarity. In any case, there are many purported roles of alpha-synuclein, and this is now mentioned in the last paragraph starting with Additionally, α-syn has been implicated…”

(5) What happened after stimulation when synapsin is detached from SV, does α-syn continues to be linked to it?

The fate of alpha-synuclein after stimulation is unclear in our experiments. Previous experiments suggest that while both synapsin and alpha-synuclein detach from the SV cluster during stimulation, synapsin returns to synapses while alpha-synuclein does not 5. However, our more recent experiments (unpublished) suggest that the activity-induced dispersion of alpha-synuclein might be phosphorylation-dependent, and that over-expression of alpha-synuclein may not be the best setting to evaluate protein dispersion. We hope to answer this question more rigorously using alpha-synuclein knock-in constructs.

(6) The experiment with E-domain fused to syPhy assumes that α-syn will still be bound to the SV. So how does α-syn inhibit ST?

The goal of this experiment was to force the synapsin E-domain to be in a location where it would normally be present – i.e. surface of the synaptic vesicle – by tagging it to sypHy (sypHy-E), and ask if this forced-retention would be sufficient to reinstate the alpha-synuclein mediated attenuation of SV-recycling (as shown in Fig. 3F, it does). Please note that the sypHy-E in these experiments does target to the synapses (new data, Fig. 3-Fig. Supp. 2D). In this context, we are not sure what the reviewer means by “So how does a-syn inhibit synaptic transmission?” We don’t think that alpha-synuclein needs to unbind from the SVs in order to inhibit synaptic transmission. Overall, we think that alpha-synuclein needs to cooperate with synapsins to perform its function, but as mentioned above and in the manuscript, the precise role of alpha-synuclein in this process is still unclear.

(7) An interesting experiment will be the expression of the isolated E-domain and examining blockage of α-syn inhibition and disruption of synapsin- α-syn interaction. Have the authors examined it as was done in other models?

We did do the experiment where we only over-expressed the isolated synapsin E-domain in neurons. We were thinking that perhaps the E-domain would have a dominant-negative effect on SV-clustering, as it did in the lamprey and other model-systems, where the E-peptide was directly injected into the axon. However, we found that in cultured hippocampal neurons, the over-expressed E-domain behaves like a soluble protein and is not enriched in synapses (see new data, Fig. 3-Fig. Supp. 2B). Also, the over-expressed E-domain cannot reinstate the synaptic attenuation induced by alpha-synuclein (new data, Fig. 3-Fig. Supp. 2C), likely because the E-domain does not target to synapses. Actually, this is why we did the syPhy-E domain experiment in the first place, to ensure that the E-domain was in the right location to have an effect.

(8) A schematic model/scheme providing a mechanistic view of the interplay between the proteins is essential and can improve the paper.

The only model we can confidently make right now would be stick-figures showing the site where alpha-synuclein C-terminus binds to synapsin, which is obviously not very insightful. As noted above (and in the revised version), several different functions have been attributed to alpha-synuclein, and the precise role of alpha-synuclein/synapsin interactions in regulating the SV-cycle is unclear. We hope to create a better model after getting some more data from us and our colleagues working on this challenging problem.

References

(1) Kononenko NL & Haucke V. (2015) Molecular mechanisms of presynaptic membrane retrieval and synaptic vesicle reformation. Neuron 85, 484-496.

(2) Gitler D, Xu Y, Kao H-T, Lin D, Lim S, Feng J, Greengard P & Augustine GJ. (2004) Molecular Determinants of Synapsin Targeting to Presynaptic Terminals. J. Neurosci. 24, 3711-3720.

(3) Logan T, Bendor J, Toupin C, Thorn K & Edwards RH. (2017) α-Synuclein promotes dilation of the exocytotic fusion pore. Nat Neurosci 20, 681-689.

(4) Nemani VM, Lu W, Berge V, Nakamura K, Onoa B, Lee MK, Chaudhry FA, Nicoll RA & Edwards RH. (2010) Increased expression of alpha-synuclein reduces neurotransmitter release by inhibiting synaptic vesicle reclustering after endocytosis. Neuron 65, 66-79.

(5) Fortin DL, Nemani VM, Voglmaier SM, Anthony MD, Ryan TA & Edwards RH. (2005) Neural activity controls the synaptic accumulation of alpha-synuclein. J Neurosci 25, 10913-10921.